# Clinical assessment of transplacental transfer of maternal SARS-CoV-2 IgG antibodies against spike and nucleocapsid proteins: A chemiluminescence microparticle immunoassay study

Elora Sharmin[1]*, Ajmain Ishaat Khan[2], Sheikh Foyez Ahmed[3]

**1** Department of Pharmacology, Faculty of Basic Science and Para Clinical Science, Bangabandhu Sheikh Mujib Medical University (BSMMU), Dhaka, Bangladesh, **2** Department of Pharmacy, Faculty of Pharmacy, University of Dhaka, Dhaka, Bangladesh, **3** Department of Cardiology, Bangabandhu Sheikh Mujib Medical University (BSMMU), Dhaka, Bangladesh

* elora.sharmin@bsmmu.edu.bd

## Abstract

Understanding the safety and efficacy of COVID-19 vaccines in pregnant women and their neonates is crucial for understanding maternal and fetal outcomes, particularly the extent of passive immunity against SARS-CoV-2 which can be imparted to the neonates. The purpose of this study was to evaluate the transplacental transfer of maternal SARS-CoV-2 IgG antibodies against the spike (S) and nucleocapsid (N) proteins to neonates and understand whether factors like maternal comorbidities, gestational weeks, and neonatal birth weight have an influence on placental transfer ratios (PTR). A total of 57 pregnant women were assessed for SARS-CoV-2-specific IgG antibodies at delivery, and corresponding antibody titers were also measured in their neonates immediately after delivery. The PTRs for anti-S and anti-N IgG were calculated, and statistical analyses were performed for identifying potential influencing factors. The mean PTR for anti-S IgG was 1.38, suggesting effective placental transfer, whereas anti-N IgG had a lower PTR of 1.13, indicating limited transfer. A strong positive correlation was observed between maternal and neonatal anti-S IgG ($r = 0.558$, $p < 0.01$), whereas maternal and neonatal anti-N IgG correlated weakly ($r = 0.402$, $p = 0.02$). No significant associations were found between PTRs and maternal age, gestational weeks, neonatal birth weight, or comorbidities. Our study inferred the efficient transfer of maternal anti-S IgG, potentially conferring early neonatal immunity against SARS-CoV-2. However, the long-term persistence of these neonatal antibodies are questionable and remains an important research prospect. This study underscored the critical role of maternal vaccination for neonatal protection and can help form evidence based rational vaccination strategies for maximum neonatal protection.

**Data availability statement:** All relevant data are within the manuscript and its Supporting information files.

**Funding:** This study was funded by a research grant from Bangabandhu Sheikh Mujib Medical University (BSMMU) under the grant number "BSMMU/2022/836", and the ethical approval proposal was given by the Institutional Review Board (IRB), BSMMU under registration number 582. The funders had no role in study design, data collection and analysis, decision to publish, or preparation of the manuscript.

**Competing interests:** The authors have declared that no competing interests exist.

## Introduction

COVID-19 is a disease which is caused by a virus called severe acute respiratory syndrome coronavirus 2 (SARS-CoV-2). It was first identified in Wuhan, China, in December 2019. The disease was officially declared as a global pandemic on March 11, 2020 by World Health Organization (WHO). From the latest data on February 23, 2025, over 7.1 million confirmed deaths have been reported worldwide. In Bangladesh, the first case was detected on March 7, 2020. SARS-CoV-2 primarily binds to angiotensin-converting enzyme 2 (ACE2) receptors in lungs and enters the host cells via the S protein [1]. A range of clinical manifestations, from asymptomatic cases to severe pneumonia, acute respiratory distress syndrome (ARDS), multi-organ failure, and even death are triggered by the disease [2]. Pregnant women, due to physiological and immunological changes, are at increased risk of severe disease, preterm labor, and poor neonatal outcomes due to COVID-19 [3]. Additionally, neonates born to SARS-CoV-2-infected mothers may experience adverse effects, like respiratory distress, fetal distress, thrombocytopenia and even death [4]. Until now, vaccines remain the most efficacious way to prevent COVID-19 in people, alongside significantly reducing the complications of the disease even if the infection does occur as well [5]. Unfortunately, there is a paucity of information regarding adverse effects of COVID-19 vaccines in Bangladesh, particularly in the population of pregnant women.

The rapid global response to the pandemic had led to accelerated development of several COVID-19 vaccines, with types ranging from mRNA-based vaccines (Pfizer-BioNTech, Moderna), inactivated virus vaccines (Sinopharm, Sinovac) and viral vector vaccines (Oxford-AstraZeneca), being disseminated worldwide. Mostly, these vaccines target the S protein of SARS-CoV-2 [6]. However, the initial clinical trials of these vaccines excluded pregnant women, creating uncertainty regarding safety, immunogenicity, and transplacental antibody transfer of these vaccines [7]. Despite this, global health organizations, namely WHO, the Centers for Disease Control and Prevention (CDC), American College of Obstetricians and Gynecologists (ACOG) and the Advisory Committee on Immunization Practices (ACIP), have recommended COVID-19 vaccination during pregnancy. In fact, Flannery et al. reported that maternal vaccination results in IgG antibodies production, which are capable of crossing the placenta and may provide neonatal passive immunity [8]. However, limited data exist regarding antibody titers, adverse effects, and long-term neonatal protection, especially in low and middle income countries like Bangladesh.

Most vaccines which are commercially available induce antibodies against the receptor-binding domain (RBD) of the S protein, which prevents viral entry into host cells [6]. The efficacy of placental antibody transfer depends on many factors, including gestational age during vaccination, maternal antibody titers, transfer mechanisms, and maternal comorbidities [9]. Flannery et al. also observed that placental transfer ratios of SARS-CoV-2 antibodies increased when a longer interval between maternal infection and delivery was present [8]. This finding aligns with another study on respiratory syncytial virus (RSV) vaccines, where a vaccination-to-delivery interval of 30 days or more significantly increased antibody transfer [10].

Interestingly, Flannery et al. reported in their study that even extreme premature delivery (<31 weeks gestation) did not affect the PTR [8].

Many COVID-19 vaccines have been approved for use in Bangladesh, including mRNA vaccines (Pfizer-BioNTech, Moderna), viral vector vaccines (Oxford-AstraZeneca), and inactivated virus vaccines (Sinopharm, Sinovac). However, a lack of data exists on vaccine safety, immunogenicity, and adverse effects in pregnant women and neonates. The Bangladesh government has authorized COVID-19 vaccination for pregnant women, but there is a paucity of surveillance data regarding vaccine-related maternal and neonatal outcomes. However, due to the unique demographic and healthcare challenges in Bangladesh, it is extremely important to perform population-specific studies in order to develop public health policies and vaccination strategies.

The aim of this study is to assess the safety and effectiveness of COVID-19 vaccines in pregnant women of Bangladesh by evaluating maternal and neonatal IgG titers, as well as adverse effects and neonatal health outcomes after delivery. Presenting this information is vital for developing guidelines for maternal immunization strategies. The findings from this study will provide valuable insights into vaccine-induced active immunity in pregnant women and passive immunity in neonates, which can help policymakers make evidence-based rational decisions for improving maternal and neonatal health outcomes against COVID-19 in Bangladesh.

## Materials and methods

### Participants' recruitment

This study was a multicenter cross sectional observational study and patient's recruitment was from 03/11/2021 to 02/05/2022 in which involved pregnant women and their infants after delivery. The recruitment process and the conduction of study was both done at BSMMU (Dhaka, Bangladesh). Informed written consent was taken from both parents for baby and for mother the patients herself.The study only included pregnant women who had known vaccination status as well as complete maternal and fetal outcome data. On the contrary, women who were vaccinated entirely (i.e., all doses) before pregnancy or postpartum and women who had pregnancies complicated by fetal aneuploidy or genetic syndromes were excluded from the study. The preliminary data collection sheet was developed through a literature review and expert consultation, then refined after a pilot study on pregnant women and newborns. Socio-demographic data and local and systemic adverse effects were collected through structured interviews and Adverse events following immunization (AEFI) forms post vaccination. Eligible patients were identified, and written informed consent was obtained during hospital admission for delivery. Adverse effects in mothers were documented using the vaccine adverse effect form from the Director General of Drug Administration (DGDA), while neonatal adverse effects were recorded post-delivery following CDC guidelines. Convenience sampling was used as the sampling method. For ethical consideration, the study was conducted after approval from the Institutional Review Board (IRB) of BSMMU.

### Data collection

Collected data included maternal demographics like age, number of children and preexisting comorbidities. Maternal laboratory tests conducted before delivery included blood sugar, hemoglobin, erythrocyte sedimentation rate (ESR) and thyroid stimulating hormone (TSH). The adverse effects of COVID-19 vaccines on mother and children were documented post-delivery. Neonatal well being of babies post-delivery were assessed through APGAR (Appearance, Pulse, Grimace, Activity, and Respiration) score. Skin condition, congenital anomaly, gestational age and weights of babies post-delivery were collected as well.

### Sample collection and analysis

After obtaining informed consent, 3 mL of venous blood was collected carefully from both the mother and newborn and blood of the newborn collected from the umbilical cord immediately after delivery following proper protocol. The blood samples were stored in 1 mg/mL EDTA-containing tubes and were preserved at 4°C for up to 1–2 weeks. No long term storage was done.

Levels of antigen-specific IgG antibodies against SARS-CoV-2 post-vaccination were measured in maternal and neo-natal serum samples using the SARS-CoV-2 IgG II Quant assay on the ARCHITECT System. This assay provided both quantitative and qualitative determination of IgG antibodies which targeted the RBD of the $S_1$ subunit of the SARS-CoV-2 spike protein. It is a two-step automated chemiluminescence microparticle immunoassay (CMIA) with a documented specificity of around 90% [11].

The two-step process minimized non-specific binding, reduced interference from human anti-mouse antibodies (HAMA), and mitigated potential high-dose hook effects like prozone effect [12]. 50 AU/mL positivity threshold was set as the cut-off value. The antibody titer was determined by collaborating with the Department of Hematology and Virology, BSMMU.BMU.

### Statistical analysis

The free trial version of IBM SPSS Statistics Version 30.0.0.0 (172) was used for performing all statistical analyses. Analysis of maternal and neonatal demographic and clinical data was performed using descriptive statistics. Means ± standard deviations (SD) was used for presenting the continuous variables and categorical variables were presented as frequencies and percentages. Correlations between maternal and neonatal IgG titers for both anti-S and anti-N antibodies was performed using Spearman's correlation coefficient. PTRs were calculated as the ratio of neonatal IgG levels to maternal IgG levels for both anti-S and anti-N antibodies. Spearman's correlation coefficient was also used to quantify the relationships between PTR and maternal/neonatal factors, such as maternal age, gestational weeks, and neonatal birth weight. Mann-Whitney U test was used to evaluate the impact of comorbidities on PTR levels. A p-value of <0.05 was considered to be statistically significant for all analyses. The correlations between maternal and neonatal IgG levels were visualized using scatter plots.

## Results

### Maternal and neonatal demographic and clinical data

Based on the inclusion criteria, a total of 57 pregnant women and their newborns were included in the study. The mothers included in the study had a mean age of 28.77 ± 7.31 years and an average gestational age at delivery of fairly normal 37.93 ± 0.84 weeks. 26.3% of the mothers had pre-existing comorbidities like gestational diabetes mellitus (GDM), hypertension, hypothyroidism, and asthma. However, no AEFIs were reported (Table 1).

As for neonatal outcomes, the babies had an average birth weight of 2.92 ± 0.39 KGs. The APGAR scores reported at 1 minute and 5 minutes after birth indicated agreeable neonatal health for most, with scores of 92.98% and 98.24% respectively. All the babies had normal skin conditions at birth with no reported congenital anomalies (Table 2).

### Correlations between maternal and neonatal anti-S and anti-N IgG titers

Spearman's correlation was used to quantify the correlations between maternal and neonatal anti-S and anti-N IgG titers. The strongest positive correlation was found between maternal and neonatal anti-N IgG titers (r = 0.742, p < 0.01). A good

**Table 1. Mothers' demographic and clinical data.**

| Age (Years) | Mean, SD | 28.77 | 7.31 |
|---|---|---|---|
| Gestational Age (Weeks) | Mean, SD | 37.93 | 0.84 |
| Number of Children (N) | Mean, SD | 1.4561 | 1.55919 |
| Comorbidities | N, % | 15 | 26.316 |
| AEFIs | N, % | 0 | 0 |

positive correlation was also reported between maternal and neonatal anti-S IgG titers (r = 0.558, p < 0.01). However, no significant relationship was observed between maternal anti-N and neonatal anti-S IgG titers as well as maternal anti-S and neonatal anti-N IgG titers, which is suggestive of the specificity of transplacental vertical transfer mechanisms (Table 3). The correlations are graphically illustrated through scatter plots from (Fig 1).

### Placental transfer ratio (PTR) analysis

The mean PTRs for anti-S and anti-N IgG are presented in Table 4. The PTR for anti-S IgG was 1.38 ± 1.77, whereas the PTR for anti-N IgG was 1.13 ± 1.01. It is inferred from these findings that anti-S IgG was more efficiently transferred across the placenta compared to anti-N IgG.

### Correlation between PTR and maternal-neonatal parameters

The correlations between PTRs and maternal/neonatal traits were performed using Spearman correlation analysis which are summarized in Table 5. No significant correlations between PTRs and maternal age, birth weight, or gestational age at delivery were seen.

The Mann-Whitney U test was performed for PTR differences in mothers with and without comorbidities which are presented in Table 6. No significant differences were observed between groups for anti-S IgG PTR (z = −0.471, p = 0.638) or anti-N IgG PTR (z = −0.1, p = 0.921), suggesting that maternal comorbidities did not influence placental transfer of antibodies.

## Discussion

The observed PTR of 1.38 for anti-S IgG is consistent with previous studies which reported efficient transplacental transfer of spike-specific antibodies following maternal vaccination or infection [13]. On the other hand, the PTR of 1.13 for anti-N IgG suggests limited transfer, consistent with studies indicating poor transfer of nucleocapsid antibodies [14]. In spite of variations in demographic data, no significant correlations were found between PTR and maternal age, birth weight, or gestational age. This is indicative that maternal physiology and placental $F_c$ receptor dynamics are key factors

**Table 2. Neonatal demographic and clinical data.**

| | | | |
|---|---|---|---|
| Body Weight (KGs) | Mean, SD | 2.92 | 0.39 |
| Congenital Anomalies | N, % | 0 | 0 |
| Normal APGAR Score at 1 minute | N, % | 53 | 92.98 |
| Normal APGAR Score at 5 minutes | N, % | 56 | 98.24 |
| Number of Male babies | N, % | 32 | 56.1 |
| Number of Female babies | N, % | 25 | 43.9 |
| Normal Skin Conditions after birth | N, % | 57 | 100 |

**Table 3. Correlations between maternal and babies anti-N and anti-S IgG titers.**

| | Maternal Anti-S IgG Titers (AU/mL) | Maternal Anti-N IgG Titers (AU/mL) |
|---|---|---|
| Neonatal Anti-S IgG Titers (AU/mL) | r = 0.558[a] p < 0.01[b] | r = 0.012 p = 0.93 |
| Neonatal Anti-N IgG Titers (AU/mL) | r = 0.402[a] p = 0.02[a] | r = 0.742 p < 0.01[b] |

[a]Significant at p < 0.05; [b]Significant at p < 0.01.

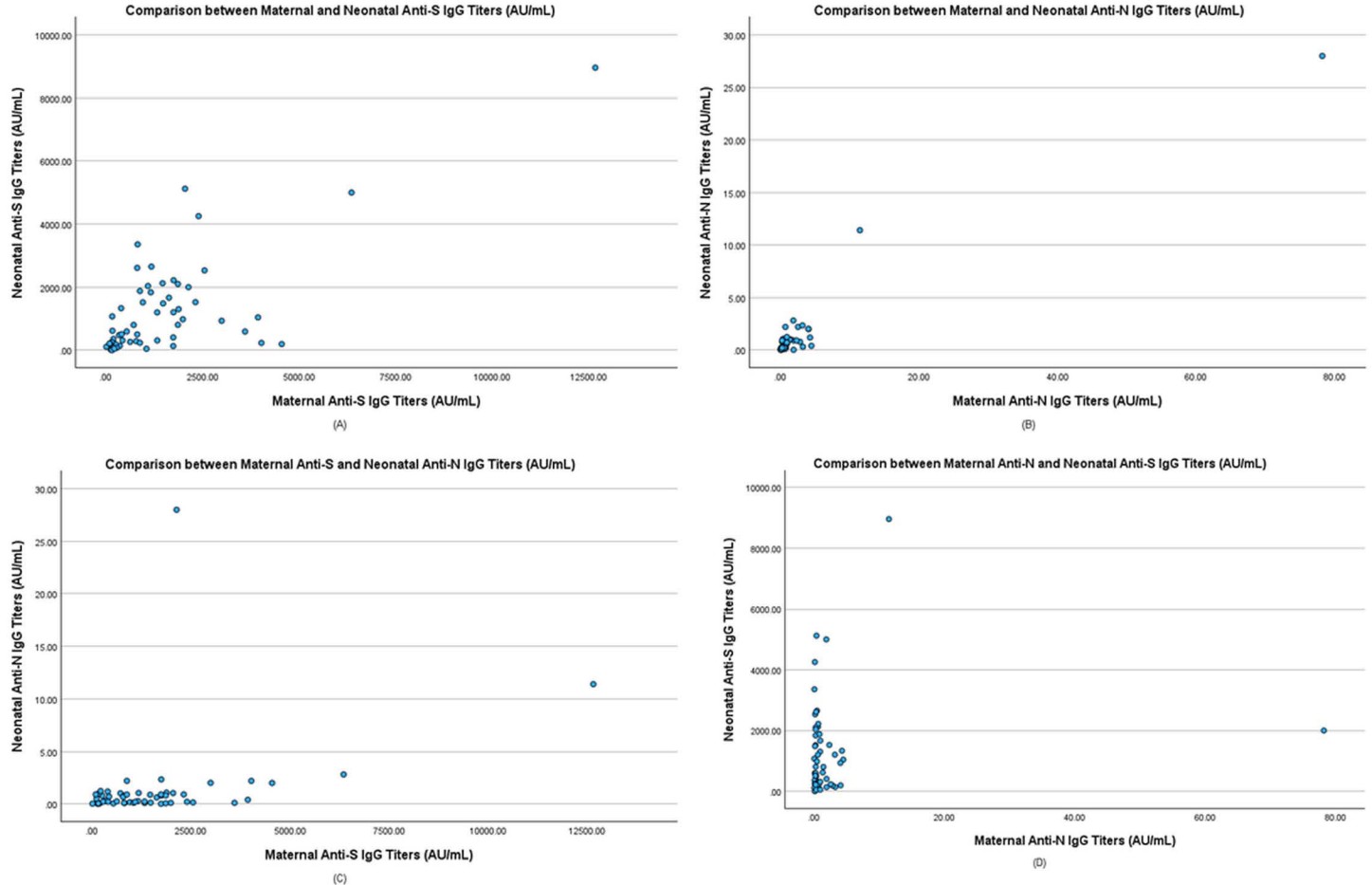

**Fig 1. Comparison between A) Maternal Anti-S and Neonatal Anti-S B) Maternal Anti-N and Neonatal Anti-N C) Maternal Anti-S and Neonatal Anti-N D) Maternal Anti-N and Neonatal Anti-S IgG titers (AU/mL).**

**Table 4. PTRs of Anti-S and Anti-N IgG.**

| PTR of Anti-S IgG | Mean, SD | 1.3842 | 1.77724 |
|---|---|---|---|
| PTR of Anti-N IgG | Mean, SD | 1.1342 | 1.01153 |

**Table 5. Correlations between PTRs and parameters using Spearman correlation.**

| | PTR of Anti-S IgG | PTR of Anti-N IgG |
|---|---|---|
| Mother's Age (years) | r=−0.178 p=0.184 | r=−0.210 p=0.117 |
| Baby's birth weight (KGs) | r=−0.192 p=0.153 | r=−0.088 p=0.513 |
| Gestational Age at delivery (weeks) | r=−0.109 p=0.419 | r=0.075 p=0.58 |

**Table 6. Correlation between PTRs and co-morbidities using Mann-Whitney U test.**

|  | PTR of Anti-S IgG | PTR of Anti-N IgG |
|---|---|---|
| Comorbidities (Yes/No) | z=−0.471<br>p=0.638 | z=−0.1<br>p=0.921 |

of transplacental IgG transfer rather than gestational and demographic factors [15]. The lack of a significant impact of maternal comorbidities on PTR is conflicting with studies of Stach et al. [16] who reported increased IgG transfer during hypertension. However, our sample size may not have been large enough to detect subtle effects, and future research has the potential to explore how chronic maternal illnesses can affect transplacental antibody transfer. The strong correlation between maternal and neonatal anti-S IgG infer that effective passive immunity against SARS-CoV-2 in neonates born to vaccinated mothers is consistent with the reports of Sunder et al. [17]. This might help provide adequate temporary protection against COVID-19 in early infancy, when neonates are particularly vulnerable to infections. However, the lifetime of these IgG antibodies is questionable and in need of additional follow up study. In contrast, the comparatively poor transfer of anti-N IgG suggests that neonates are less likely to acquire nucleocapsid-mediated immunity, limiting the diagnostic utility of neonatal anti-N IgG in differentiating maternal infection from neonatal exposure. This finding has significant implications for elucidating neonatal serological results in further epidemiological studies.

Vaccination against COVID-19 reportedly produced antibodies against both N and S proteins of the SARS-CoV-2 virus and has been reported for long term immunity also [18]. This study provided important insights into the transplacental transfer of maternal SARS-CoV-2 antibodies, selectively emphasizing between anti-S and anti-N IgG titers. From our findings, it can be inferred without doubt that vaccinating pregnant mothers against SARS-CoV-2 is an effective way to protect neonates against the disease, which is consistent with the study of Sunder et al. [17] and Partey et al. [19] as well. Moreover, Flannery et al. [8] also reported that the antibody titers after vaccination were at least 10 fold higher than the titers levels after infection. Although this study doesn't explore it, it can be hypothesized that the PTRs of anti-N and anti-S IgG titers will also be higher in vaccination compared to infection and it remains an important research prospect.

## Conclusion

Delineating the safety and effectiveness of COVID-19 vaccines in pregnant women and their neonates after delivery is crucial for understanding whether the vaccines produce any adverse effects in the pregnant mother and fetus, as well as if they could impart adequate passive immunity in the neonates against COVID-19 or not. The parameter used here in order to understand this was the transplacental transfer of maternal SARS-CoV-2 IgG antibodies for both S and N proteins. Additionally the effect of potential influencing factors such as maternal comorbidities, gestational age, and neonatal birth weight on PTRs were also assessed. Our study found that maternal anti-S IgG was effectively transferred across the placenta, with a mean PTR of 1.38, whereas anti-N IgG showed a lower transfer efficiency, with a PTR of 1.13. Although, a strong positive correlation (r=0.742, p<0.01) existed between maternal and neonatal anti-N IgG titers compared maternal anti-S IgG with neonatal anti-S IgG (r=0.558, p<0.01), reinforcing the hypothesis that antibodies are selectively transferred. No significant correlation was found between PTRs and maternal age, gestational weeks, neonatal birth weights as well as comorbidities. These findings corresponded with previous studies which demonstrated that anti-S IgG is preferentially transferred in comparison to anti-N IgG. It can be confidently inferred that neonates born to vaccinated or previously infected mothers receive robust spike-specific passive immunity, which may provide early protection against SARS-CoV-2 infection. Despite these valuable insights, our study had limitations. The sample size, if larger, could have resulted in a more robust statistical inference. Moreover, we couldn't check the effect of many other factors such as BMI, vaccination timing, etc on PTRs. Furthermore, we were unable to conduct any follow-up studies to determine the persistence of neonatal IgG titers months after birth. Finally, the study was conducted in a single geographic region, which may limit the speculation of the findings to other populations with different

ethnicities, vaccination rates, comorbidity profiles, or healthcare systems. Future studies should follow-up on and investigate the persistence of maternally derived SARS-CoV-2 antibodies in neonates and their effectiveness against infections. A much more deeper insight can be explored in elucidating why S proteins had a larger transfer ratio than the N proteins. Large-scale multicenter studies incorporating diverse ethnicity and maternal health conditions could further clarify the impact of comorbidities on IgG transfer efficiency. In the end, the effect of booster vaccination strategies during pregnancy and its effect on neonatal immunity could be explored for further optimization of the established vaccination protocol for pregnant mothers. This study emphasized on the vital role of maternal vaccination in imparting passive immunity to neonates, prioritizing the need for continued maternal immunization in order to safeguard neonates against SARS-CoV-2 infection. By further understanding the mechanisms mediating transplacental antibody transfer, it is possible to develop targeted maternal vaccination approaches that can maximize neonatal protection against future pandemics and infectious disease outbreaks.

## Supporting information

**S1 File. Data sheet.**
(XLSX)

## Acknowledgments

The authors are grateful to the Department of Hematology and Virology, BSMMU for performing the necessary immunoassays required for this study.

## Author contributions

**Conceptualization:** Elora Sharmin, Sheikh Foyez Ahmed.

**Data curation:** Ajmain Ishaat Khan.

**Formal analysis:** Ajmain Ishaat Khan.

**Funding acquisition:** Elora Sharmin, Sheikh Foyez Ahmed.

**Investigation:** Elora Sharmin.

**Methodology:** Elora Sharmin, Sheikh Foyez Ahmed.

**Project administration:** Elora Sharmin.

**Resources:** Elora Sharmin.

**Software:** Ajmain Ishaat Khan.

**Validation:** Elora Sharmin.

**Visualization:** Ajmain Ishaat Khan.

**Writing – original draft:** Ajmain Ishaat Khan.

**Writing – review & editing:** Elora Sharmin, Sheikh Foyez Ahmed.

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
