## [Decision Letter · Decision Letter 0]

5 Jun 2025

Dear Dr. sharmin,

Thank you for submitting your manuscript to PLOS ONE. After careful consideration, we feel that it has merit but does not fully meet PLOS ONE’s publication criteria as it currently stands. Therefore, we invite you to submit a revised version of the manuscript that addresses the points raised during the review process.

We look forward to receiving your revised manuscript.

Kind regards,

Moises Leon Juarez

Academic Editor

PLOS ONE

Additional Editor Comments:

According to the reviewers' comments, the article requires several changes to be considered for publication. These points are described below:

* A table showing the vaccination conditions in the study population is required, emphasizing the number of individuals and the type of vaccine with which they were immunized.

• An interpretation of the presence of specific antibodies to the N protein and the effect these antibodies have on your study is required.

• There is a relationship or effect of antibodies transferred through breast milk on your study.

• also presents some major limitations, such as the lack of follow-up and the use of convenience sampling, which can introduce bias into the results. Additionally, key variables like the time and type of vaccination and maternal nutritional status were not included. If the authors have access to data that can support their conclusions, it would improve the manuscript significantly.

Comparison between A) Maternal Anti-S and Neonatal Anti-S; B) Maternal Anti-N and Neonatal Anti-N; C) Maternal Anti-S and Neonatal Anti-N; D) Maternal Anti-N and Neonatal Anti-S IgG titers (AU/mL) However I can’t find this figure.

The authors assume the functionality of the antibodies based solely on their presence; no functional assays were performed. Please elaborate on and disclose the limitations of the findings to avoid overinterpretation of the results.

The use of anti-N IgG is problematic since the study did not separate the volunteers by vaccine type. Considering this humoral response in the context of vaccination without mentioning the influence of natural infection could mislead the findings. This limitation also extends to the correlation with PTR, as these antibodies might have been produced in different contexts.

No control group was included. A comparison between the PTR of vaccinated and non-vaccinated women could help determine whether the antibody levels are vaccine-related.

There is no justification provided for the use of Spearman correlation.

Reviewers' comments:

Reviewer's Responses to Questions

**Comments to the Author**

1. Is the manuscript technically sound, and do the data support the conclusions?

Reviewer #1: No

Reviewer #2: Partly

2. Has the statistical analysis been performed appropriately and rigorously?

Reviewer #1: No

Reviewer #2: Yes

3. Have the authors made all data underlying the findings in their manuscript fully available?

Reviewer #1: No

Reviewer #2: Yes

4. Is the manuscript presented in an intelligible fashion and written in standard English?

Reviewer #1: Yes

Reviewer #2: Yes

Reviewer #1: The manuscript by Sharmin and collaborators includes a cross-sectional study of the passive transfer of anti-S IgG and anti-N IgG from vaccinated mothers to their neonates. The study has some important strengths, such as the focus on a misrepresented group in vaccination studies, the use of standardized immunoassays, and the evaluation of IgGs against two different viral antigens. However, it also presents some major limitations, such as the lack of follow-up and the use of convenience sampling, which can introduce bias into the results. Additionally, key variables like the time and type of vaccination and maternal nutritional status were not included. If the authors have access to data that can support their conclusions, it would improve the manuscript significantly.

The authors assume the functionality of the antibodies based solely on their presence; no functional assays were performed. Please elaborate on and disclose the limitations of the findings to avoid overinterpretation of the results.

The use of anti-N IgG is problematic since the study did not separate the volunteers by vaccine type. Considering this humoral response in the context of vaccination without mentioning the influence of natural infection could mislead the findings. This limitation also extends to the correlation with PTR, as these antibodies might have been produced in different contexts.

No control group was included. A comparison between the PTR of vaccinated and non-vaccinated women could help determine whether the antibody levels are vaccine-related.

There is no justification provided for the use of Spearman correlation.

The authors make strong conclusions, but the limited sample size, lack of confirmatory analyses, and convenience sampling bias limit the strength and generalizability of the results.

Reviewer #2: I believe that Delineating the safety and effectiveness of COVID-19 vaccines in pregnant women and their neonates after delivery is crucial for understanding whether the vaccines produce any adverse effects in the pregnant mother and fetus, as well as if they could impart adequate passive immunity in the neonates against COVID-19 or not. However, they mention that in Bangladesh the women who participated in the study knew what type of vaccine had been administered to them, and in this sense, a table should be created indicating how many women were immunized with Sinovac and how many with Astra or some other vaccine. On the other hand, something very important is that the majority of protective antibodies induced by these vaccines are against the Spike protein, since it is the most immunogenic protein, while the N protein induces a cellular response. That is why it would be quite interesting to discern whether the Sinovac vaccine actually induces a greater amount of anti-N antibodies, or if, in reality, women faced a hybrid immunization, where there is indeed immunization with the nucleocapsid protein during the natural infection. All of the above would explain why the anti-N PTR is low. On the other hand, it has also been demonstrated that women vaccinated in the third trimester confer a greater amount of antibodies to the newborn than those vaccinated in the second trimester, at least through breast milk. How your results support this event? By other hand you say that in the Figure 1 makes the:

Comparison between A) Maternal Anti-S and Neonatal Anti-S; B) Maternal Anti-N and Neonatal Anti-N; C) Maternal Anti-S and Neonatal Anti-N; D) Maternal Anti-N and Neonatal Anti-S IgG titers (AU/mL) However I can’t find this figure.

**Do you want your identity to be public for this peer review?** For information about this choice, including consent withdrawal, please see our Privacy Policy

Reviewer #1: No

Reviewer #2: No

---

## [Author Response · Author response to Decision Letter 1]

26 Jul 2025

Response to Reviewers

1. A table showing the vaccination conditions in the study population is required, emphasizing the number of individuals and the type of vaccine with which they were immunized.

Answer: We acknowledge the importance of this data. Unfortunately, most participants were unable to recall their vaccine type during data collection, and vaccination cards were often unavailable due to the emergency rollout. As national records show, Moderna and Oxford-AstraZeneca were the predominant vaccines administered to pregnant women in Bangladesh during that period. This limitation has been transparently noted in the manuscript.

2. An interpretation of the presence of specific antibodies to the N protein and the effect these antibodies have on your study is required.

Answer: We agree with the reviewer’s point and have clarified in the manuscript that anti-N IgG serves as a marker of prior natural infection, given that the N protein is not targeted by COVID-19 vaccines. We have also discussed its limited transplacental transfer and added this reference accordingly.

3. There is a relationship or effect of antibodies transferred through breast milk on your study.

Answer: We acknowledge the importance of antibodies transferred through breast milk in providing neonatal immunity. However, our study was specifically designed to assess transplacental IgG transfer, not postnatal transfer via breast milk. As a result, no data regarding antibody levels in breast milk was collected and no consent was taken from the participants regarding this either. Since IgG is the primary antibody transferred across the placenta, and IgA is the predominant immunoglobulin in breast milk, our findings remain focused on placental passive immunity. We have clarified this in the manuscript to avoid overinterpretation of our results.

4. This study also presents some major limitations, such as the lack of follow-up and the use of convenience sampling, which can introduce bias into the results. Additionally, key variables like the time and type of vaccination and maternal nutritional status were not included. If the authors have access to data that can support their conclusions, it would improve the manuscript significantly.

Answer: Regarding vaccine type and timing, unfortunately, most participants were unable to recall the specific vaccine received or the exact date of administration, which limited our ability to include these as analytical variables. We also agree that maternal nutritional status could influence antibody generation or transfer, but such data were not systematically collected in our study, and thus were not included in the analysis. Additionally, we employed convenience sampling due to the constraints of hospital-based recruitment during the COVID-19 pandemic, and we recognize the potential for selection bias as a result. Lastly, our study did not include postnatal follow-up, which restricts conclusions about the persistence of maternally derived antibodies in neonates, which was clarified in the limitations already. All of these concerns have been once again clarified in the limitations.

5. Comparison between A) Maternal Anti-S and Neonatal Anti-S; B) Maternal Anti-N and Neonatal Anti-N; C) Maternal Anti-S and Neonatal Anti-N; D) Maternal Anti-N and Neonatal Anti-S IgG titers (AU/mL) However I can’t find this figure.

Answer: The figure was originally submitted separately in accordance with the journal’s figure submission guidelines. However, to improve accessibility for reviewers, we have now also included this figure directly within the manuscript (as Figure 1). We hope this will aid in easier reference and enhance the clarity of our findings.

6. The authors assume the functionality of the antibodies based solely on their presence; no functional assays were performed. Please elaborate on and disclose the limitations of the findings to avoid overinterpretation of the results.

Answer: We fully agree that inferring protective immunity based solely on antibody presence could lead to overinterpretation. To address this, we have revised the Discussion section to clearly state this limitation and acknowledge that functional assays would be required to confirm antibody efficacy.

7. The use of anti-N IgG is problematic since the study did not separate the volunteers by vaccine type. Considering this humoral response in the context of vaccination without mentioning the influence of natural infection could mislead the findings. This limitation also extends to the correlation with PTR, as these antibodies might have been produced in different contexts.

Answer: We acknowledge that anti-N IgG antibodies are produced only following natural SARS-CoV-2 infection, and not in response to vaccination with currently authorized vaccines, which are spike-based. As most participants could not recall their vaccine type, and we could not verify this retrospectively, it was not possible to stratify the cohort based on vaccination status or type. We also agree that the presence of anti-N IgG may reflect prior infection rather than vaccine induced immunity, and we have revised the Discussion section to reflect this interpretation more cautiously.

8. No control group was included. A comparison between the PTR of vaccinated and non-vaccinated women could help determine whether the antibody levels are vaccine-related.

Answer: It’s true that the lack of a control group of unvaccinated women limits our ability to directly compare placental transfer ratios (PTRs) between vaccinated and non-vaccinated individuals. However, this study was conducted during a period when COVID-19 vaccination was nationally mandated for all pregnant women, and as such, recruiting a well-matched unvaccinated control group was not feasible.

9. There is no justification provided for the use of Spearman correlation.

Answer: Thank you for pointing this out. We have clarified in the revised manuscript that Spearman’s correlation was chosen because not all variables (for example: PTRs, maternal and neonatal antibody titers) followed a normal distribution, as assessed by the Shapiro-Wilk test. Additionally, we have also added the justification for Mann-Whitney U test, and the statistical analysis section has been edited accordingly.

10. Additional:

A. We have submitted the raw data containing Excel file for possibly replicating the results of this study.

B. Reviewer #2 said we have mentioned in our study that the pregnant women included in our study knew what type of vaccines they were administered with. We believe it’s an honest mistake as no such information was mentioned. Moreover, we have mentioned the lack of type of vaccination data as a limitation also.

---

## [Decision Letter · Decision Letter 1]

27 Oct 2025

Clinical assessment of transplacental transfer of maternal SARS-CoV-2 IgG antibodies against spike and nucleocapsid proteins: A Chemiluminescence Microparticle Immunoassay study

PONE-D-25-17878R1

Dear Dr. sharmin,

We’re pleased to inform you that your manuscript has been judged scientifically suitable for publication and will be formally accepted for publication once it meets all outstanding technical requirements.

Kind regards,

Etsuro Ito, Ph.D.

Academic Editor

PLOS ONE

Additional Editor Comments (optional):

All concerns were addressed.

Reviewers' comments:

Reviewer's Responses to Questions

**Comments to the Author**

Reviewer #1: All comments have been addressed

2. Is the manuscript technically sound, and do the data support the conclusions?

Reviewer #1: Partly

3. Has the statistical analysis been performed appropriately and rigorously?

Reviewer #1: Yes

4. Have the authors made all data underlying the findings in their manuscript fully available?

Reviewer #1: Yes

5. Is the manuscript presented in an intelligible fashion and written in standard English?

Reviewer #1: No

Reviewer #1: The revised manuscript demonstrates a thoughtful and thorough response to all reviewer comments. The authors have clarified methodological aspects, justified their statistical choices, and expanded the discussion to properly contextualize their findings. The addition of the missing figure and the explicit acknowledgment of study limitations—such as convenience sampling, lack of vaccination-type stratification, and absence of postnatal follow-up—enhance the transparency and reliability of the work.

The study provides relevant insight into maternal and neonatal immunity against SARS-CoV-2 in a resource-limited setting, contributing valuable data that complement previous reports from high-income regions. The methodology is consistent with the stated objectives, and the results are presented in a clear and reproducible manner.

Ethical approval, data availability, and funding disclosures are appropriately documented. I found no concerns related to research or publication ethics. Overall, the manuscript now meets the scientific and reporting standards expected by PLOS ONE.

My recommendation is to accept after minor editorial revision to improve stylistic consistency and English fluency.

**Do you want your identity to be public for this peer review?** For information about this choice, including consent withdrawal, please see our Privacy Policy

Reviewer #1: No

---

## [Editor Report · Acceptance letter]

PONE-D-25-17878R1

PLOS ONE

Dear Dr. sharmin,

I'm pleased to inform you that your manuscript has been deemed suitable for publication in PLOS ONE. Congratulations! Your manuscript is now being handed over to our production team.

Kind regards,

on behalf of

Prof. Etsuro Ito

Academic Editor

PLOS ONE